# Corporate Social Responsibility and Product Market Power

**Chong-Chuo Chang** [1] , **Han Yang** [2,*] **and Kun-Zhan Hsu** [1]

1    Department of Banking and Finance, College of Management, National Chi Nan University, Nantou 54561, Taiwan; aaron@ncnu.edu.tw (C.-C.C.); s104214048@mail1.ncnu.edu.tw (K.-Z.H.)
2    Department of Strategy and Development of Emerging Industries, College of Management, National Chi Nan University, Nantou 54561, Taiwan
*    Correspondence: aprilyanghan@gmail.com; Tel.: +86-13739484801

**Abstract:** This study explores the impact of corporate social responsibility (CSR) on the product market power by examining listed firms on the Taiwan Stock Exchange and Taipei Exchange from 2005 to 2017. We use CSR awards as a social responsibility indicator, and the results show a positive relationship between CSR and excess price-cost margins (market share), supporting the thesis that firms that value CSR activities can strengthen the competitive advantage of products in the market.

**Keywords:** corporate social responsibility; product market power; excess price-cost margin; market share





## 1. Introduction

Corporate social responsibility (CSR) is advocated as a key component of the social contract between businesses and society (Fernández-Kranz and Santaló 2010). The idea of CSR is becoming increasingly important in today's business climate, as firms actively pursue economic growth through internationalization[1] (Feng et al. 2018). CSR has gradually become an issue of concern among business managers, as firms are evaluated not only on financial performance but also on their social image (Rhou et al. 2016).

The Business Council for Sustainable Development of Taiwan (BCSD-Taiwan, ROC) officially joined the World Business Council for Sustainable Development (WBCSD) in 1997, shortly after both were founded. The objective was to follow international trends, hold seminars, and conduct research projects that promote the sustainable development of Taiwanese firms. Taiwan's *CommonWealth Magazine* took the lead in advocating the importance of corporate social responsibility (CSR) in 1997 and established the "Excellence in Corporate Social Responsibility" award (hereinafter referred to as "ECSR") to examine the companies' responsibilities to shareholders, employees, partners, society, and the environment from four major aspects: corporate governance, corporate commitment, social participation, and environmental protection. *Global Views Monthly* also established the "Corporate Social Responsibility Award" (hereinafter referred to as "CSRA") and "Social Enterprise Award" (hereinafter referred to as "SEA"), which aim to motivate firms to create a more positive influence on their surroundings and society. Through the promotion of the magazines' investigations and reporting, CSR is no longer an unfamiliar concept but one that has gradually come to affect the choices of investors and consumers. Furthermore, the KPMG Survey of Corporate Responsibility Reporting 2017 reported that CSR reporting rate increased by 11 percentage points in Taiwan between 2015 and 2017. Simultaneously, Taiwan became one of the ten countries with the highest rates of CSR information in annual financial reports.

The formal definition of CSR has been discussed across multiple domestic and foreign forums, but it remains inconsistent. Carroll (1979) defines four types of CSR: economic, legal, ethical, and discretionary. Jamali (2008) emphasizes that CSR has always been described as an elusive, complex, and fuzzy concept. According to the WBCSD, CSR is a

commitment to the achievement of a high-quality life for employees, families, communities, localities, and society, to achieve sustainable economic development.

Some studies have investigated the impact of CSR on financial performance (Cochran and Wood 1984; Ramzan et al. 2021; Tsoutsoura 2004; Uyar et al. 2020; Saeidi et al. 2015; Peloza 2009; Madime and Gonçalves 2022), firm value (Gregory et al. 2014; Qiu et al. 2020), corporate innovation (Gallego-Álvarez et al. 2011; Chkir et al. 2020), and reputation (Eberle et al. 2013; Fombrun and Shanley 1990; Fombrun 2005; Sánchez-Torné et al. 2020). However, few studies have examined the relationship between CSR and product market power. Datta et al. (2013) revealed that strong product market power allows for greater flexibility in responding to unexpected changes in consumer product needs. Furthermore, strong market power is associated with a more stable cash flow and lower stock return volatility (Peress 2010) and improved stock liquidity (Kale and Loon 2011; Acharya and Pedersen 2005; Tookes 2008).

Previous research indicates that CSR performance can establish a positive image (Pomering and Johnson 2009; Du et al. 2010), bolster firms' reputations (Sánchez-Torné et al. 2020; Fombrun and Shanley 1990; Dhaliwal et al. 2012), and provide competitive advantage in the market (Fry et al. 1982; Rhou et al. 2016; Porter and Kramer 2002). Participation in CSR activities can not only enhance the firms' stock value (Anderson and Frankle 1980; Flammer 2013; Klassen and McLaughlin 1996; Porter and Kramer 2002) but may also further impact firms' sales growth (Lins et al. 2017). In addition, Gonçalves et al. (2021b) clear that green funds (funds formed by companies with high environmental, social, and governance (ESG) scores, or even funds specialized in sustainable, non-polluting, and environmentally friendly companies) exhibit higher risk-adjusted returns during crisis periods.

Firms can also build their images, strengthen relationships with stakeholders, and enhance stakeholder advocacy behaviors by engaging in CSR activities (Balmer and Greyser 2006; Salmones et al. 2005; Du et al. 2010). Therefore, this study believes that focusing on CSR will help firms increase product market power.

To investigate the relationship between CSR and product market power, we study a sample of listed firms from Taiwan from 2005 to 2017, using CSR awards as a social responsibility indicator. The results indicate that firms with CSR awards exhibit higher excess price-cost margin and market share, which implies that firms can employ CSR activities to increase their competitive advantage in the product marketplace. Furthermore, related results also indicate that firms' awards-winning level has a significant positive impact on excess price-cost margin.

The remainder of this paper is organized as follows: Section 2 contains the literature review and hypotheses development; Section 3 describes the sample and introduces the methodology, including the research models and the calculation and measurement of variables; Section 4 discusses the empirical results of the study; Section 5 considers the endogenous problem; and Section 6 concludes the paper.

## 2. Literature Review and Hypotheses Development

An increasing number of marketing communicators incorporate the firms' CSR information into their brand marketing because previous research works have suggested that good CSR performance can establish a positive image (Pomering and Johnson 2009; Du et al. 2010), bolster reputation for firms, and improve their competitive advantage in the market. CSR initiatives have often served businesses as sources of competitive advantage (Rhou et al. 2016; Porter and Kramer 2002). Fry et al. (1982) suggest that effective CSR strategies can help gain active support, facilitate subconscious advertising, and motivate firms to engage in competitive markets. The stakeholder theory suggests that CSR plays a significant role in attaining competitive advantage, enhancing corporate reputation, and increasing firm value (Fombrun and Shanley 1990; Jamali 2008; Dhaliwal et al. 2012). Sánchez-Torné et al. (2020) also state that firms with positive levels of CSR show equally positive levels of corporate reputation. Furthermore, Han and Lee (2021) point out that financial results can benefit through the improvement of corporate image caused by increased market

share, rate of sales, and decreased operation expenses along with consequentially enhanced market value and competitiveness. These arguments provide evidence that encourages CSR involvement among listed firms.

Chang et al. (2014) indicate that stock analysts in Taiwan tend to recommend firms with superior CSR performance as investment targets and suggest holding shares in these firms (Anderson and Frankle 1980; Flammer 2013; Klassen and McLaughlin 1996; Porter and Kramer 2002). The stocks of these firms are less risky, yet demonstrate high-yield characteristics for investors (Eccles et al. 2012; Luo and Bhattacharya 2009; Waddock and Graves 1997). Gonçalves et al. (2021a) demonstrate that firms with better CSR performance report more transparent financial information and thus have higher-quality financial reporting.

From the perspective of attracting consumers, CSR has been recognized as one of the most important marketing activities to maintain good relationships with consumers (Balmer and Greyser 2006). Salmones et al. (2005) indicate that the firms' socially responsible behavior has a direct and positive influence on consumer loyalty. Furthermore, Du et al. (2010) state that CSR activities can generate positive stakeholder attitudes and support behaviors. It is beneficial for firms to build an image, strengthen relationships between stakeholders and company, and enhance stakeholder advocacy behaviors. Furthermore, Mohr et al. (2001) point out that most of the interviewees were found to expect fairly high levels of CSR, and consumers are more likely to boycott the products and services of irresponsible firms. Therefore, firms with poor CSR performance may face consumer resistance and lose market competitiveness advantage.

It is also beneficial for firms to strengthen relationships between stakeholders and company, gain consumer loyalty, and enhance advocacy behaviors, which further improve their competitiveness in the market.

Therefore, we establish the following hypotheses:

**Hypothesis 1.** *Good CSR performance helps increase excess price-cost margin.*

**Hypothesis 2.** *Good CSR performance helps increase product market share.*

### 3. Data and Methodology

*3.1. Data*

In this study, we use data from firms listed on the two major Taiwanese stock exchanges —Taiwan Stock Exchange (TWSE) and Taipei Exchange (TPE$_X$)—as samples for our research. We apply stock trading information and financial statements of sample firms obtained from the Taiwan Economics Journal for the period 2005–2017. The data were not resampled, and data with missing values were removed.

Our sample includes 2013 firms and 18,253 firm-year observations. Table 1 shows that the electronic parts industry has the highest number of firms (242), accounting for 12.02% of the total, followed by the optoelectronic industry (9.99%). Electronic parts, semiconductor, and optoelectronic industries exhibit the highest, second highest, and third highest firm-year observations (2442, 1672, and 1562, respectively) and account for 13.38%, 9.16%, and 8.56% of the total sample, respectively. Semiconductor, optoelectronic, and communication and internet industries have more CSR awards, including "ECSR" announced by *CommonWealth Magazine* (*CSR_CW* = 1), and "CSRA" or "SEA", announced by *Global Views Monthly* (*CSR_GV* = 1), with a higher number of firms as well. Automobile, trading and consumer goods, oil, gas and electricity, all receive more CSR awards, despite having a lower number of firms.

**Table 1.** Sample distribution and the industry situation of obtaining CSR awards. This table presents the sample distribution and the industry situation in obtaining CSR awards. *CSR_CW* is set at 1 if sample firms win the "ECSR" announced by *CommonWealth Magazine*; otherwise, it is set at 0. *CSR_GV* is set at 1 if sample firms win the "CSRA" or "SEA" announced by *Global Views Monthly*; otherwise, it is set at 0.

| Industry | Firm-Year Observations | Percentage | Firms | Percentage | CSR_CW = 1 | Percentage | CSR_GV = 1 | Percentage |
|---|---|---|---|---|---|---|---|---|
| Cement | 99 | 0.54% | 8 | 0.40% | 5 | 1.06% | 1 | 0.67% |
| Food | 333 | 1.82% | 31 | 1.54% | 8 | 1.70% | 4 | 2.67% |
| Plastic | 333 | 1.82% | 28 | 1.39% | 3 | 0.64% | 2 | 1.33% |
| Textile | 705 | 3.86% | 61 | 3.03% | 23 | 4.88% | 5 | 3.33% |
| Electric Machinery | 975 | 5.34% | 112 | 5.56% | 23 | 4.88% | 2 | 1.33% |
| Electrical and Cable | 201 | 1.10% | 18 | 0.89% | 0 | 0.00% | 0 | 0.00% |
| Glass and Ceramic | 59 | 0.32% | 6 | 0.30% | 0 | 0.00% | 0 | 0.00% |
| Paper and Pulp | 91 | 0.50% | 7 | 0.35% | 6 | 1.27% | 0 | 0.00% |
| Iron and Steel | 530 | 2.90% | 47 | 2.33% | 10 | 2.12% | 0 | 0.00% |
| Rubber | 149 | 0.82% | 13 | 0.65% | 1 | 0.21% | 0 | 0.00% |
| Automobile | 65 | 0.36% | 5 | 0.25% | 29 | 6.16% | 9 | 6.00% |
| Building Material and Construction | 950 | 5.20% | 83 | 4.12% | 9 | 1.91% | 3 | 2.00% |
| Shipping and Transportation | 303 | 1.66% | 32 | 1.59% | 5 | 1.06% | 1 | 0.67% |
| Tourism | 292 | 1.60% | 39 | 1.94% | 19 | 4.03% | 10 | 6.67% |
| Trading and Consumers' Goods | 327 | 1.79% | 32 | 1.59% | 36 | 7.64% | 15 | 10.00% |
| Other | 950 | 5.20% | 100 | 4.97% | 11 | 2.34% | 0 | 0.00% |
| Chemical | 515 | 2.82% | 51 | 2.53% | 10 | 2.12% | 0 | 0.00% |
| Biotechnology and Medical Care | 914 | 5.01% | 139 | 6.91% | 0 | 0.00% | 0 | 0.00% |
| Oil, Gas and Electricity | 165 | 0.90% | 14 | 0.70% | 52 | 11.04% | 24 | 16.00% |
| Semiconductor | 1672 | 9.16% | 191 | 9.49% | 90 | 19.11% | 17 | 11.33% |
| Computer and Peripheral Equipment | 1298 | 7.11% | 123 | 6.11% | 16 | 3.40% | 8 | 5.33% |
| Optoelectronic | 1562 | 8.56% | 201 | 9.99% | 58 | 12.31% | 24 | 16.00% |
| Communication and Internet | 948 | 5.19% | 104 | 5.17% | 34 | 7.22% | 17 | 11.33% |
| Electronic Parts | 2442 | 13.38% | 242 | 12.02% | 3 | 0.64% | 3 | 2.00% |
| Electronic Products | 554 | 3.04% | 52 | 2.58% | 5 | 1.06% | 2 | 1.33% |
| Information Service | 428 | 2.34% | 44 | 2.19% | 15 | 3.18% | 3 | 2.00% |
| Other Electronic | 941 | 5.16% | 98 | 4.87% | 0 | 0.00% | 0 | 0.00% |
| Others | 452 | 2.48% | 132 | 6.56% | 0 | 0.00% | 0 | 0.00% |
| Total | 18,253 | 100.00% | 2013 | 100.00% | 471 | 100.00% | 150 | 100.00% |

*3.2. Methodology*

To investigate whether firms that improve their involvement in CSR activities can increase product market power, this study establishes the following regression model:

$$MarketPower_{it} = \alpha_i + \beta_1 CSR_{it-1} + \sum_{n=1}^{N} \delta_n CV_{nit-1} + \eta_t + \varepsilon_{it} \qquad (1)$$

where *i* denotes the firm, and *t* denotes the year. *MarketPower*$_{it}$ is the product market power variable of the *i*-th sample firm in *t* year. Brine et al. (2007) note that it may also be useful to use a one-year lag between the measurement of financial performance and the corporate social responsibility measure to determine whether there may be a lag associated with the implementation of social responsibility and improved financial performance (Blackburn et al. 1994). Thus, considering that there may be a lag between the firm obtaining CSR awards and product market power, we also lag the product market power variables by one period. $CSR_{it-1}$ is the CSR variable of the *i*-th sample firm in *t*−1 year; $CV_{nit}$ is the *n*-th sample firm in *t* − 1 year, indicating value of control variables. Referring to Datta

et al. (2013) and Amidu and Wolfe (2013), the project includes variables of firm size (*SIZE*), long-term debt ratio (*LTD*), sales growth rate (*SG*), and market-to-book ratio (*MB*) to control their impact on product market power. Considering the anomalies of years and industries, we further include the industry dummy variable (*IndDummies*) and year dummy variable (*YearDummies*) in the regression model. Year dummy variables are included to control the impact of the macroeconomic environment in different years on product market power, such as the impact of the 2008–2009 global financial crisis and the 2011–2012 European debt crisis.

### 3.3. Empirical Variables

3.3.1. Product Market Power

Most studies use one of the EPCM and MS to measure product market power, such as Kale and Shahrur (2007), Peress (2010), Kale and Loon (2011), Datta et al. (2013). In order to obtain more robust results, we use two measures of a firm's product market power: the excess price-cost margin (*EPCM*) and market share (*MS*). The formulae of measurement are as follows:

$$PCM_{it} = \frac{Sales_{it} - COGS_{it} - SG\&A_{it}}{Sales_{it}} \tag{2}$$

$$EPCM_{it} = PCM_{it} - IndPCM_{it} \tag{3}$$

$$MS_{it} = \frac{Sales_{it}}{\sum_{j=1}^{K} Sales_{ijt}} \tag{4}$$

In Equations (2) and (3), *Sales* indicates net sales; *COGS* indicates cost of goods sold; *SG&A* indicates sales, general, and administrative expenses, and *PCM* indicates price-cost margin generated from *Sales* excluding *COGS* and *SG&A*, divided by *Sales*. The *EPCM* is the difference between the PCM of the sample firm (*PCM*) and the PCM of industry (*IndPCM*)[2]. The higher the *EPCM*, the more capable the firm is to set prices at marginal cost. Thus, firm products are more competitive in the market. In Equation (4), *K* is the number of firms in the corresponding industry. *MS* is the net sales of the sample firm, divided by the total net sales of the sample firm's industry. The higher the MS, the more competitive the firm's products are in the market.

3.3.2. CSR Variables

In this study, we use CSR awards, conferred by Taiwanese magazines as social responsibility indicators. Therefore, we set two dummy variables (*CSR_CW* and *CSR_GV*) to represent the CSR awards. *CSR_CW* is set at 1 if sample firms win the "ECSR" announced by the *CommonWealth Magazine*; otherwise, it is set at 0. *CSR_GV* is set at 1 if sample firms win the "CSRA" or "SEA" announced by *Global Views Monthly*; otherwise, it is set at 0.

3.3.3. Control Variables

The firm size (*SIZE*) is the natural logarithm of the total assets of the sample firm; the long-term debt ratio (*LTD*) is calculated by dividing the long-term liabilities by total assets; the sales growth rate (*SG*) is the difference between the sales of the current year and the sales of the previous year, divided by sales of previous year. The market-to-book ratio (*MB*) is the ratio of market value of common stock divided by the net value of shareholders' equity.

## 4. Empirical Results

### 4.1. Difference in Corporate Performance between Firms That Do and Do Not Obtain Awards

Table 2 illustrates the difference in corporate performance between firms that do and do not obtain "ECSR". All firm-year observations are classified into two groups, based on the achievement (*CSR_CW* = 1) or forfeiture (*CSR_CW* = 0) of the *CommonWealth Magazine's* award "ECSR". We compare the difference in the mean and median between the two samples. The results indicate that firms with a CSR award (*CSR_CW* = 1) exhibit higher mean and median values for excess price-cost margin (*EPCM*), market share (*MS*), firm

size (*SIZE*), and market-to-book ratio (*MB*), all reaching the significance level of 1%. The results support the finding that firms with CSR awards exhibit higher product market power. For example, the difference in the *EPCM* mean (median) between award-winning firms and firms that did not win awards is 0.0932 (0.0321), both values being statistically significant at a 1% significance level. The difference of the *MS* mean (median) between award-winning firms and firms that did not win awards is 0.0314 (0.0184), both values being statistically significant at the 1% significance level. The *SIZE* and *MB* also display the mean (median) between award-winning firms and firms that did not obtain awards as 2.0729 (1.8592) and 0.5781 (0.6003), respectively, all values being statistically significant at the 1% significance level.

**Table 2.** Difference in product market power between firms with and without *CommonWealth Magazine's* CSR award. This table presents the differences in product market power between firms with and without "ECSR", as announced by *CommonWealth Magazine*. Differences in the mean and median are assessed using the *t*-test and the Wilcoxon rank sum test. *** and * represent 1% and 10% significance levels, respectively.

| Panel A. Mean Difference | | | | |
|---|---|---|---|---|
| **Variable** | ***CSR_CW* = 1** | ***CSR_CW* = 0** | **Difference** | ***p*-Value** |
| *EPCM* | 0.0548 | −0.0383 | 0.0932 *** | <0.0001 |
| *MS* | 0.0434 | 0.0120 | 0.0314 *** | <0.0001 |
| *SIZE* | 16.6546 | 14.5817 | 2.0729 *** | <0.0001 |
| *LTD* | 0.0620 | 0.0613 | 0.0007 | 0.8848 |
| *SG* | 0.0386 | 0.0628 | −0.0242 * | 0.0524 |
| *MB* | 2.1110 | 1.5329 | 0.5781 *** | <0.0001 |
| Panel B. Median Difference | | | | |
| **Variable** | ***CSR_CW* = 1** | ***CSR_CW* = 0** | **Difference** | ***p*-Value** |
| *EPCM* | 0.0309 | −0.0012 | 0.0321 *** | <0.0001 |
| *MS* | 0.0214 | 0.0031 | 0.0184 *** | <0.0001 |
| *SIZE* | 16.6484 | 14.7892 | 1.8592 *** | <0.0001 |
| *LTD* | 0.0001 | 0.0103 | −0.0101 | 0.8291 |
| *SG* | 0.0255 | 0.0001 | 0.0254 * | 0.0972 |
| *MB* | 1.8294 | 1.2291 | 0.6003 *** | <0.0001 |

Table 3 illustrates the difference in corporate performance between firms that do and do not obtain "ECSR". In this case, all firm-year observations are classified into two groups, based on firms receiving (*CSR_GV* = 1) or forfeiting (*CSR_GV* = 0) the "CSRA" or "SEA" award announced by *Global Views Monthly*. We compare the difference in the mean and median between the two samples. These findings indicate that firms that obtained the CSR awards exhibited higher mean and median values for excess price-cost margin (*EPCM*), market share (*MS*), firm size (*SIZE*), and market-to-book ratio (*MB*), all values attaining significant levels. The results suggest that firms obtaining CSR awards exhibit higher product market power. For example, the difference in the *EPCM* mean (median) between award-winning firms and firms that did not win awards is 0.0829 (0.0182), both values being statistically significant at the 1% significance level. The difference of the *MS* mean (median) between award-winning firms and firms that did not win awards is 0.0501 (0.0326), both values being statistically significant at the 1% significance level. Additionally, the *SIZE* and *MB* also display the mean (median) between award-winning firms and firms that did not win awards, with 2.5635 (3.0477), 1.0853 (0.9888), respectively, all values being statistically significant at the 1% significance level.

**Table 3.** Difference in product market power between firms with and without *Global Views Monthly's* CSR awards. This table presents the differences in product market power between firms with and without "Corporate Social Responsibility" award or "SEA," as announced by *Global Views Monthly*. Differences in the mean and median are assessed using the *t*-test and the Wilcoxon rank sum test. ***, **, and * represent 1%, 5%, and 10% significance levels, respectively.

| Panel A. Mean Difference | | | | |
|---|---|---|---|---|
| **Variable** | **CSR_GV = 1** | **CSR_GV = 0** | **Difference** | ***p*-Value** |
| *EPCM* | 0.0485 | −0.0344 | 0.0829 *** | <0.0001 |
| *MS* | 0.0632 | 0.0130 | 0.0501 *** | <0.0001 |
| *SIZE* | 17.1317 | 14.5681 | 2.5635 *** | <0.0001 |
| *LTD* | 0.0791 | 0.0637 | 0.0155 | 0.1223 |
| *SG* | 0.0257 | 0.0736 | −0.0479 ** | 0.0184 |
| *MB* | 2.5985 | 1.5132 | 1.0853 *** | <0.0001 |
| **Panel B. Median Difference** | | | | |
| **Variable** | **CSR_GV = 1** | **CSR_GV = 0** | **Difference** | ***p*-Value** |
| *EPCM* | 0.0175 | −0.0007 | 0.0182 *** | <0.0001 |
| *MS* | 0.0359 | 0.0034 | 0.0326 *** | <0.0001 |
| *SIZE* | 17.8471 | 14.7994 | 3.0477 *** | <0.0001 |
| *LTD* | 0.0472 | 0.0140 | 0.0332 * | 0.0883 |
| *SG* | 0.0448 | 0.0109 | 0.0339 | 0.7416 |
| *MB* | 2.2045 | 1.2156 | 0.9888 *** | <0.0001 |

*4.2. Regression Results*

Table 4 shows the relationship between firms whether obtain "ECSR" award from *CommonWealth Magazine* and the product market power. The excess price-cost margin (*EPCM*) and the market share (*MS*) are the independent variables. The dependent variable *CSR_CW* is set at 1 if sample firms win the "ECSR", as announced by *CommonWealth Magazine*; otherwise, it is set at 0. The empirical results demonstrate that the *CSR_CW* for all regression models exhibit significantly positive relationships with *EPCM* and *MS*, at the 1% significance level. The empirical results support Hypotheses 1 and 2, demonstrating that good CSR performance helps increase firms' excess price-cost margins and market share. By participating in CSR-related activities, firms can establish a positive image, bolster reputation, and gain competitive advantage in the market, thereby increasing the excess price-cost margin and market share. Therefore, firms can increase their product market power by participating in CSR activities and try to improve the probability of obtaining CSR awards.

In terms of the control variables, firm size (*SIZE*) exhibits a significantly positive relationship with *EPCM* and *MS*, implying that larger firms exhibit higher product market power. The long-term debt ratio (*LTD*) exhibits significantly different relationships with the two product market power variables. In terms of *EPCM*, the results indicate that the *LTD* exhibits significantly negative relationships with the *EPCM*, at the 1% significance level. However, the *LTD* exhibits a significantly positive relationship with *MS*, at the 1% significance level. Moreover, the coefficients of sales growth rate (*SG*) and market-to-book ratio (*MB*) are significantly positive at the 1% significance level with only *EPCM*, suggesting that higher sales growth rates and market-to-book ratios can acquire the advantage of pricing.

Table 5 shows the relationship between firms whether obtain "CSRA" or "SEA" from *Global Views Monthly* and the product market power. The independent variables are the excess price-cost margin (*EPCM*) and the market share (*MS*). The dependent variable, *CSR_GV*, is set at 1 if sample firms win the "CSRA" or "SEA" announced by *Global Views Monthly*; otherwise, it is set at 0. The empirical results demonstrate that the *CSR_GV* for all regression models exhibit significantly positive relationships with *EPCM* and *MS*, at the 1% significance level, indicating that firms with CSR awards have higher product market power as compared to firms without CSR awards. Again, the results support Hypotheses 1 and 2.

**Table 4.** Relationship between Firms whether Obtain "ECSR" Award of *CommonWealth Magazine* and Product Market Power. This table presents results for the pooled ordinary least squares (OLS) method estimation for the relationship between firms that do or do not win "ECSR", as announced by *CommonWealth Magazine*, and product market power. The figure in brackets is the standard error of the Newey–West correction self-correlation and heterogeneous variability (Newey and West 1987). *** and ** represent 1% and 5% significance levels, respectively.

| | (1) | (2) | (3) | (4) |
|---|---|---|---|---|
| | *EPCM* | *MS* | *EPCM* | *MS* |
| *Intercept* | −0.2095 *** | −0.0228 *** | −0.2451 *** | −0.0139 *** |
| | (0.0219) | (0.0019) | (0.0251) | (0.0021) |
| $CSR\_CW_{t-1}$ | 0.0586 *** | 0.0280 *** | 0.0595 *** | 0.0230 *** |
| | (0.0065) | (0.0033) | (0.0070) | (0.0026) |
| $SIZE_{t-1}$ | 0.0115 *** | 0.0024 *** | 0.0092 *** | 0.0021 *** |
| | (0.0015) | (0.0001) | (0.0014) | (0.0001) |
| $LTD_{t-1}$ | −0.1547 *** | 0.0122 *** | −0.1306 *** | 0.0094 *** |
| | (0.0230) | (0.0027) | (0.0243) | (0.0023) |
| $SG_{t-1}$ | 0.0198 *** | 0.0005 | 0.0214 *** | 0.0001 |
| | (0.0071) | (0.0003) | (0.0074) | (0.0003) |
| $MB_{t-1}$ | 0.0079 *** | −0.0001 | 0.0153 *** | 0.0004 ** |
| | (0.0027) | (0.0002) | (0.0028) | (0.0002) |
| *Industry dummies* | | | Yes | Yes |
| *Year dummies* | | | Yes | Yes |
| *Adjusted* $R^2$ | 0.0217 | 0.0776 | 0.0432 | 0.3745 |
| *F-value* | 67.35 *** | 252.48 *** | 17.48 *** | 219.30 *** |

**Table 5.** Relationship between Firms whether Obtain "CSRA" or "SEA" of *Global Views Monthly* and Product Market Power. This table presents the pooled ordinary least squares (OLS) method estimation results for the relationship between firms that win or do not win the "CSRA" or "SEA", as announced by *Global Views Monthly*, and product market power. The figure in brackets is the standard error of the Newey–West correction self-correlation and heterogeneous variability (Newey and West 1987). ***, **, and * represent 1%, 5%, and 10% significance levels, respectively.

| | (1) | (2) | (3) | (4) |
|---|---|---|---|---|
| | *EPCM* | *MS* | *EPCM* | *MS* |
| *Intercept* | −0.1867 *** | −0.0230 *** | −0.2328 *** | −0.0142 *** |
| | (0.0178) | (0.0016) | (0.0217) | (0.0019) |
| $CSR\_GV_{t-1}$ | 0.0494 *** | 0.0442 *** | 0.0478 *** | 0.0317 *** |
| | (0.0105) | (0.0075) | (0.0119) | (0.0060) |
| $SIZE_{t-1}$ | 0.0099 *** | 0.0025 *** | 0.0079 *** | 0.0021 *** |
| | (0.0012) | (0.0001) | (0.0012) | (0.0001) |
| $LTD_{t-1}$ | −0.1379 *** | 0.0130 *** | −0.1203 *** | 0.0108 *** |
| | (0.0206) | (0.0026) | (0.0217) | (0.0022) |
| $SG_{t-1}$ | 0.0185 ** | 0.0010 *** | 0.0189 ** | 0.0006 |
| | (0.0075) | (0.0004) | (0.0078) | (0.0004) |
| $MB_{t-1}$ | 0.0103 *** | −0.0005 ** | 0.0173 *** | 0.0004 * |
| | (0.0025) | (0.0002) | (0.0026) | (0.0002) |
| *Industry dummies* | | | Yes | Yes |
| *Year dummies* | | | Yes | Yes |
| *Adjusted* $R^2$ | 0.0192 | 0.0725 | 0.0397 | 0.3648 |
| *F-value* | 68.80 *** | 271.54 *** | 17.65 *** | 232.26 *** |

In terms of the control variables, firm size (*SIZE*) exhibits a significantly positive relationship with *EPCM* and *MS*, implying that larger firms exhibit higher product market power. The long-term debt ratio (*LTD*) exhibits significant relationships with the two-product market power variables. In terms of *EPCM*, the results indicate that *LTD* exhibits

significantly negative relationships with it at the 1% significance level. However, *LTD* exhibits significantly positive relationships with *MS*, also at the 1% significance level. Moreover, the coefficients of the sales growth rate (*SG*) and market-to-book ratio (*MB*) are significantly positive at the 1% significance level with *EPCM*, suggesting that higher sales growth rate and market-to-book ratio can acquire the advantage of pricing.

*4.3. Results of Further Analysis*

4.3.1. Impact of Firms Obtaining CSR Awards on Product Market Power

To further explore the impact of firms obtaining CSR awards on product market power, we restrict the samples to firms with CSR awards. In Table 6, the independent variables are the excess price-cost margin (*EPCM*) and the market share (*MS*). The dependent variable *CSR_CW* is set at 1 if sample firms win the "ECSR" announced by *CommonWealth Magazine*; otherwise, it is set at 0. *CSR_GV* is set at 1 if sample firms win the "CSRA" or "SEA" announced by *Global Views Monthly*; otherwise, it is set at 0. After restricting the samples to firms with CSR awards, the results of the *CSR_CW* (*CSR_GV*) exhibit significantly positive relationships with *EPCM* at the 5% (1%) significance level, suggesting that after firms obtain CSR awards, their price-cost margins rise. Therefore, participating in CSR activities is beneficial for acquiring price advantages. Table 6 also indicates that *MB* exhibits a significantly positive relationship with both *EPCM* and *MS* at the 5% significance level or higher, suggesting that firms with higher market-to-book ratio possess higher product market power.

**Table 6.** Impact of aftermath of firms obtaining CSR awards on product market power. In this table, we restrict the samples to firms with CSR awards. The figure in brackets is the standard error of the Newey–West correction self-correlation and heterogeneous variability (Newey and West 1987). \*\*\*, \*\*, and \* represent 1%, 5%, and 10% significance levels, respectively.

| | (1) | (2) | (3) | (4) |
|---|---|---|---|---|
| | *EPCM* | *MS* | *EPCM* | *MS* |
| *Intercept* | 0.0281 | −0.6487 *** | −0.1813 *** | −0.6045 *** |
| | (0.0336) | (0.0407) | (0.0429) | (0.0523) |
| $CSR\_CW_{t-1}$ | 0.0151 ** | 0.0006 | | |
| | (0.0072) | (0.0064) | | |
| $CSR\_GV_{t-1}$ | | | 0.0267 *** | −0.0069 |
| | | | (0.0080) | (0.0077) |
| $SIZE_{t-1}$ | −0.0027 | 0.0436 *** | 0.0101 *** | 0.0405 *** |
| | (0.0019) | (0.0026) | (0.0027) | (0.0033) |
| $LTD_{t-1}$ | −0.0807 *** | −0.0319 | −0.1298 *** | −0.0083 |
| | (0.0265) | (0.0207) | (0.0374) | (0.0321) |
| $SG_{t-1}$ | 0.0065 | 0.0012 | 0.0379 * | 0.0255 |
| | (0.0098) | (0.0035) | (0.0202) | (0.0159) |
| $MB_{t-1}$ | 0.0136 *** | 0.0144 *** | 0.0164 *** | 0.0045 * |
| | (0.0019) | (0.0018) | (0.0022) | (0.0024) |
| *Industry dummies* | Yes | Yes | Yes | Yes |
| *Year dummies* | Yes | Yes | Yes | Yes |
| *Adjusted $R^2$* | 0.2038 | 0.6057 | 0.3602 | 0.6794 |
| *F-value* | 16.91 | 96.47 | 18.06 | 65.21 |

4.3.2. Impact of Award-Winning Level on Product Market Power

"ECSR" is announced by the *CommonWealth Magazine*, which scores firms based on four comprehensive indicators: corporate governance, corporate commitment, social participation, and environmental sustainability. "CSRA" and "SEA" announced by *Global Views Monthly* contains different levels of award; thus, we set two award-winning level variables: *CSR_CWScore* and *CSR_GVScore*. They aim to explore the impact of award-winning level on product market power for all firms with CSR awards. *CSR_CWScore* is the score obtained by firms from the "ECSR", announced by *CommonWealth Magazine*;

*CSR_GVScore* represents the award level obtained by firms from "CSRA" and "SEA", announced by *Global Views Monthly*, expressed through the scores of 3, 2, or 1. The firms that obtain a first prize receive scores of 3; another firm receiving a model prize receives a score of 2; and the others acquiring five-star awards receive a score of 1.

Table 7 shows that impact of award-winning level on product market power. The empirical results demonstrate that the *CSR_CWScore* (*CSR_GVScore*) for regression models exhibit a significantly positive impact on *EPCM* at the 5% (1%) significance level, proving that firms that obtain higher-level CSR awards will experience higher excess price-cost margin. Therefore, firms should aim to obtain higher-level CSR awards to gain product market power.

**Table 7.** Impact of CSR award-winning level on product market power. This table presents the impact of CSR award-winning level on product market power for all sample firms with CSR awards. The figure in brackets is the standard error of the Newey–West correction self-correlation and heterogeneous variability (Newey and West 1987). ***, **, and * represent 1%, 5%, and 10% significance levels, respectively.

| | (1) | (2) | (3) | (4) |
|---|---|---|---|---|
| | *EPCM* | *MS* | *EPCM* | *MS* |
| *Intercept* | 0.0747 | −0.7454 *** | −0.2692 *** | −0.7046 *** |
| | (0.0469) | (0.0573) | (0.0566) | (0.0698) |
| $CSR\_CWScore_{t-1}$ | 0.0024 ** | −0.0001 | | |
| | (0.0010) | (0.0008) | | |
| $CSR\_GVScore_{t-1}$ | | | 0.0088 *** | 0.0010 |
| | | | (0.0030) | (0.0026) |
| $SIZE_{t-1}$ | −0.0040 * | 0.0481 *** | 0.0157 *** | 0.0453 *** |
| | (0.0023) | (0.0035) | (0.0033) | (0.0039) |
| $LTD_{t-1}$ | −0.1167 *** | −0.0727 ** | −0.2383 *** | −0.0065 |
| | (0.0401) | (0.0286) | (0.0435) | (0.0349) |
| $SG_{t-1}$ | 0.0041 | −0.0025 | 0.0149 | 0.0435 ** |
| | (0.0100) | (0.0031) | (0.0182) | (0.0194) |
| $MB_{t-1}$ | 0.0196 *** | 0.0167 *** | 0.0195 *** | 0.0067 ** |
| | (0.0032) | (0.0028) | (0.0026) | (0.0030) |
| *Industry dummies* | Yes | Yes | Yes | Yes |
| *Year dummies* | Yes | Yes | Yes | Yes |
| *Adjusted $R^2$* | 0.2429 | 0.6010 | 0.4190 | 0.7243 |
| *F-value* | 12.92 | 56.95 | 15.06 | 52.23 |

## 5. Endogeneity

### 5.1. Panel Data Regression

We adopt a panel data regression to verify the relationship between firms that obtain CSR awards and their product market power. The regression results in Table 8 present a positive relationship between $CSR\_CW_{t-1}$ and *EPCM* at the 1% level; the $CSR\_GV_{t-1}$ also has a significantly positive impact on *MS*. Therefore, our results support Hypotheses 1 and 2.

### 5.2. Generalized Method of Moments

We apply the GMM methodology of Arellano and Bond (1991) to resolve the endogeneity problems. Table 9 presents the GMM estimation results for the relationship between firms that obtain CSR awards and their product market power. The empirical results demonstrate that *CSR_CW* and *CSR_GV* are both positively related to the *EPCM* and *MS* at the 1% significance level, suggesting that firms that obtain CSR awards can exercise higher product market power.

**Table 8.** Relationship between firms that obtain CSR awards and product market power: panel data regression. This table presents the panel data regression results for the relationship between firms that obtain CSR awards and their product market power for all sample firms. The figure in brackets is the standard error of the Newey–West correction self-correlation and heterogeneous variability (Newey and West 1987). *** and ** represent 1% and 5% significance levels, respectively.

| | CSR Awards of *CommonWealth Magazine* | | CSR Awards of *Global Views Monthly* | |
|---|---|---|---|---|
| | *EPCM* | *MS* | *EPCM* | *MS* |
| *Intercept* | −1.2072 *** | −0.0554 *** | −1.2936 *** | −0.0610 *** |
| | (0.2557) | (0.0032) | (0.0785) | (0.0029) |
| $CSR\_CW_{t-1}$ | 0.0230 *** | −0.0007 | | |
| | (0.0045) | (0.0007) | | |
| $CSR\_GV_{t-1}$ | | | 0.0037 | 0.0023 ** |
| | | | (0.0423) | (0.0011) |
| $SIZE_{t-1}$ | 0.0763 *** | 0.0045 *** | 0.0822 *** | 0.0049 *** |
| | (0.0173) | (0.0002) | (0.0053) | (0.0002) |
| $LTD_{t-1}$ | −0.0992 ** | −0.0043 *** | −0.1155 *** | −0.0056 *** |
| | (0.0404) | (0.0011) | (0.0283) | (0.0011) |
| $SG_{t-1}$ | 0.0554 *** | 0.0006 *** | 0.0431 *** | 0.0008 *** |
| | (0.0077) | (0.0001) | (0.0035) | (0.0001) |
| $MB_{t-1}$ | 0.0122 *** | 0.0006 *** | 0.0149 *** | 0.0005 *** |
| | (0.0035) | (0.0001) | (0.0022) | (0.0001) |
| *Firm dummies* | Yes | Yes | Yes | Yes |
| *Year dummies* | Yes | Yes | Yes | Yes |
| *Adjusted $R^2$* | 0.5707 | 0.9329 | 0.5450 | 0.9253 |
| *F-value* | 11.5492 *** | 111.2113 *** | 11.3542 *** | 108.3268 *** |

**Table 9.** Relationship between firms that obtain CSR awards and product market power: GMM estimation. This table presents the GMM estimation results for the relationship between firms that obtain CSR awards and product market power. Standard errors are reported in parentheses. The figure in brackets is the standard error of the Newey–West correction self-correlation and heterogeneous variability (Newey and West 1987). ***, **, and * represent 1%, 5%, and 10% significance levels, respectively.

| | CSR Awards of *CommonWealth Magazine* | | CSR Awards of *Global Views Monthly* | |
|---|---|---|---|---|
| | *EPCM* | *MS* | *EPCM* | *MS* |
| *Intercept* | −0.0088 *** | −0.0004 ** | −0.0092 *** | −0.0002 ** |
| | (0.0019) | (0.0002) | (0.0012) | (0.0001) |
| $CSR\_CW_{t-1}$ | 1.3101 *** | 0.1886 *** | | |
| | (0.1090) | (0.0092) | | |
| $CSR\_GV_{t-1}$ | | | 0.8155 *** | 0.1687 *** |
| | | | (0.1661) | (0.0120) |
| $SIZE_{t-1}$ | −0.0068 *** | 0.0013 *** | −0.0021 *** | 0.0020 *** |
| | (0.0011) | (0.0001) | (0.0007) | (0.0001) |
| $LTD_{t-1}$ | −0.0792 ** | 0.0100 *** | −0.0921 *** | 0.0083 *** |
| | (0.0343) | (0.0029) | (0.0277) | (0.0020) |
| $SG_{t-1}$ | 0.0262 *** | 0.0006 | 0.0156 *** | 0.0009 ** |
| | (0.0059) | (0.0005) | (0.0049) | (0.0004) |
| $MB_{t-1}$ | −0.0050 * | −0.0005 * | 0.0037 | −0.0002 |
| | (0.0029) | (0.0002) | (0.0024) | (0.0002) |
| *Industry dummies* | Yes | Yes | Yes | Yes |
| *Year dummies* | Yes | Yes | Yes | Yes |

## 6. Conclusions

CSR has become increasingly important in the current business climate, as firms positively pursue economic growth through internationalization. However, most of the existing literature on corporate governance has discussed issues regarding the relationship between CSR activities and financial performance. Previous studies have seldom explored issues regarding improvement of product market power; hence, aiming to conduct extensive research about CSR, we conducted an empirical analysis of the relationship between CSR and product market power. Thus, we tried to determine whether firms' focus on CSR activities serves as a source of competitive advantage or not.

Our sample consisted of 2013 firms and 18,253 firm-year observations for the period of 2005–2017. We used CSR awards as a social responsibility indicator, and the empirical results show that firms with CSR awards exhibit higher excess price-cost margin and market share. This finding indicates that firms can profit financially and socially from CSR activities to increase the competitive advantage of their products in the marketplace. Moreover, CSR award-winning level also exhibits significantly positive impact on excess price-cost margins. Our study contributes to the understanding of the relationship between CSR and product market power of firms, which consequently encourages firms to try and pay better attention to CSR activities. Therefore, we conclude that strengthening the idea of corporate social responsibility and participating in CSR activities can help firms improve the competitive advantage of their products in the market.

There are two main limitations that need to be acknowledged and addressed in the present study. The first limitation is that not all firms in Taiwan will be evaluated by CSR awards. Therefore, more measures of CSR can be added in the future research. In addition, ESG can also be considered in this study as a set of standards used by potential investors to screen firms they could potentially invest in, which means that exploring the relationship between ESG and product market power is also an interesting direction for future research.

**Author Contributions:** Conceptualization, C.-C.C. and K.-Z.H.; methodology, C.-C.C., H.Y. and K.-Z.H.; software, C.-C.C. and K.-Z.H.; formal analysis, C.-C.C., H.Y. and K.-Z.H.; investigation, C.-C.C., H.Y. and K.-Z.H.; data curation, C.-C.C. and K.-Z.H.; writing—original draft preparation, C.-C.C., H.Y. and K.-Z.H.; writing—review and editing, C.-C.C. and H.Y.; visualization, C.-C.C. and H.Y.; project administration, C.-C.C., H.Y. and K.-Z.H.; funding acquisition, C.-C.C. All authors have read and agreed to the published version of the manuscript.

**Funding:** Chong-Chuo Chang gratefully acknowledges financial support from the Ministry of Science and Technology of Taiwan: MOST 110-2410-H-260-006-MY3.

**Conflicts of Interest:** The authors declare no conflict of interest.

## Notes

[1]　BynoteJanuary 201, more than 60 countries and 3000 firms and organizations had adopted the Global Reporting Initiative and made commitments to compile continuous development reports.

[2]　The price-cost margin of industry (*IndPCM*) is based on the weighted average calculated by the price-cost margin of sample firms' market share multiplied by the price-cost margin of firms.

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
