# Peer review of "Corporate Social Responsibility and Product Market Power"

_economies, doi:10.3390/economies10060151_

Round 1

Reviewer 1 Report

Dear authors, 

We find the research interesting and important. However, we would like to highlight the following improvement opportunities:
 - Line 52-53: "However, However" -> "However"
 - Line 81: "2. Hypotheses Development" -> "2. Literature review and Hypotheses Development"
 - Line 82: "More and more marketing communicators take firms’ CSR" -> "An increasing number of marketing communicators take firms’ CSR"
 - Line 120: "Therefore, we establish Hypothesis 1 as follows:" -> "Therefore, we establish the following hypotheses:"
 - Line 125-128: Please provide further details regarding the data: granularity of data obtained, was the data resampled?, are there any missing values?, how do the authors acknowledge and consider any anomalies from the 2008-2009 Global Financial Crisis? are there other anomalies to be considered? what granularity do the authors consider when computing the regression?
 - Line 130-141: Please provide further details on why and how the selected variables are important to the market power: SIZE, LTD, SG, MB, ...
 - Line 144: "Product market power" -> this seems to be dangling without context
 - Line 145-147: Provide further details why the choice of EPCM and MS is best to assess market power and their relationship to CSR. Are there other alternatives? Why do the authors pick these and not others?
 - Line 175: "Sample description" -> remove. 
 - Line 176-193 + Table 1: they should be included in section "3.1 Data".
 - Line 195: "Difference in corporate performance between firms that do and don’t obtain awards." -> is this a sub-sub-sub-section?
 - Line 196-210: The authors evaluate the results mentioning statistical levels of significance. May we ask for further details? Which statistical test was applied / which null hypothesis was tested? How many records were considered?

 Tables:
  - Table 1: "Sample Distribution and the Industry Situation of Obtaining CSR Awards. Table 1. if..." -> "Table 1" should be removed from the description. Shorten the overall description.
  - Table 2: Shorten the overall description. Make sure the boldening of the table title is consistent (see: "Panel A. Mean difference", "Panel B. Median difference")
  - Table 3: Shorten the overall description. Make sure the boldening of the table title is consistent (see: "Panel A. Mean difference", "Panel B. Median difference")
  - Table 4, 5, 6, 7: Shorten the overall description.

Other observations:
 - Please use some acronyms for the following awards, that are mentioned through the text: "Excellence in Corporate Social Responsibility", "Corporate Social Responsibility Award", and "Social Enterprise Award".
 - The authors do not discuss the fact that there may be some delay between the attribution of some CSR awards and the perceived market power. Is there some literature or knowledge, that could provide insights on this? How can affect the conclusions?
 - In all tables: highlight the first-best and second-best results using bold and italics. We suggest aligning the numerical results to the right while aligning texts to the left.

Reviewer 2 Report

Dear authors

I thoroughly enjoyed your paper and I find the topic and conclusions have room to impact extant literature and managerial implications.

Yet, You should consider the folllowing notes:

1 - The doccument requires extensive editing, proof reading and improvement in terms of english (both grammar and syntax)

2 - Your research design does not consider issues related to endogeneity (better performing firms, both in terms of CSR and Product market power) will accumulate positive excess performance in the future. As such, i would reccomend that you employ a more robust analysis based on panel data econometrics, either fixed or random effects, depending on Hausman tests.

3 - Furthermore, it would be relevant to look at the issues under analysis at dynamic perspective - using for instance a GMM methodology.

4 - Finally, please explain what you mean with "virtual variables" (pag. 4).

Kind regards 

Reviewer 3 Report

The paper must be very extensively re-written. It requires EXTENSIVE editing from beginning to end.

P. 4: clarify the precise meaning of "net sales" and SG&A"

p. 5: is SG a percentage?

pp. 8-9 and thereafter. Why used pooled dat? Why not estimate using Random Effects or Fixed Effects after applying the Hausman test? The results could be interesting if the estimation technique were undertaken accordingly.

Round 2

Reviewer 1 Report

Dear authors, 

We thank you for your response. We have examined the responses and the manuscript and would like to point to some improvement opportunities.

 - (1) 3.2. Methodology: how do the year dummy variables control the influence of special years (e.g., regarding the Global Financial Crisis)? We would appreciate some evidence in that regard. Furthermore, most of the questions posed regarding the data, remained unanswered: the granularity of data obtained, was the data resampled?, are there any missing values?, how do the authors acknowledge and consider any anomalies from the 2008-2009 Global Financial Crisis? are there other anomalies to be considered? what granularity do the authors consider when computing the regression?

 - (2) Response 8: we appreciate the answer. The authors cite many more related works in the answer than in the manuscript. Please, update accordingly, also letting the reader know that "EPCM and MS are commonly used as a proxy variable for power firm's product market power. Most studies use one of the EPCM and MS to measure product market power".

 - (3) Response 18: the original question remains unanswered: The authors do not discuss the fact that there may be some delay between the attribution of some CSR awards and the perceived market power. Is there some literature or knowledge, that could provide insights on this? How can affect the conclusions? Is the decision to consider a single period arbitrary?

 - (4) Conclusion: please enhance the conclusions, by highlighting some limitations and the scope of the approach described in this research.

Reviewer 2 Report

Dear Authors,

Thank you for your revision of the paper.

It seems you have mixed my notes on your repply. Please refer to it and provide a comment on note number 4.

Additionally, i miss references from the journal or mdpi journals, at least. Examples may include:

Madime, E., & Gonçalves, T. C. (2022). Consequences of Social and Environmental Corporate Responsibility Practices: Managers’ Perception in Mozambique. International Journal of Financial Studies10(1), 4.

Gonçalves, T., Pimentel, D., & Gaio, C. (2021). Risk and performance of European green and conventional funds. Sustainability13(8), 4226.

Gonçalves, T., Gaio, C., & Ferro, A. (2021). Corporate social responsibility and earnings management: Moderating impact of economic cycles and financial performance. Sustainability13(17), 9969.

Kind regards

Reviewer 3 Report

NA

Author Response

Thanks for your comments and encouragement. We believe that the manuscript has been greatly improved by these revisions, and we hope that you will now find it suitable for publication in Economies

Round 3

Reviewer 1 Report

Dear authors, 

We appreciate the changes introduced into the manuscript. Nevertheless, we would like to encourage you to consider the following observations:

(1)- In Section 3.1 include the following details:

  - The data was not resampled 

  - Data with missing values has been removed, so there is no problem with missing values.

(2)- In Section 3.2 include the following details (rewrite as appropriate):

  - "Year dummy variables are included to control the impact of the macroeconomic environment in different years on product market power. Thus, it not only controls the impact of the 2008-2009 Global Financial Crisis but also controls the impact of the 2011-2012 European Debt Crisis."

  - "We mainly consider the anomalies of years and industries, so we included the industry dummy variable (IndDummies) and year dummy variable (YearDummies) in the regression model. Year dummy variable not only controls the impact of the 2008-2009 Global Financial Crisis but also controls the impact of the 2011-2012 European Debt Crisis."

(3)- The authors should include the following explanation in the paper: "The lag of CSR is also considered in some scientific literature, so a one-period lag is used to reduce the endogeneity problem. For example, Brine, Brown & Hackett (2007) note that it may also be useful to use a one year lag between the measurement of financial performance and the corporate social responsibility measure to determine whether there may be a lag associated with the implementation of social responsibility and improved financial performance (Blackburn, Doran and Shrader 1994). Thus, considering a lag between the firms obtaining CSR awards and product market power, we also lag the product market power variables by one period."

(4)- The Market Power equation in Section 3.2 is unreadable. Please update.

(5)- Table 3, 4, 5, 6, 7, 8, 9: The descriptions are too long. Please move part of the explanation into the text and keep just what is strictly necessary to correctly understand the Table's information.

(6)- Table 2: The authors mention that "***, **, and * represent 1%, 5%, and 10% significance levels", but only results at 1% and 10% significance levels are shown. Is this an error? Please consider reviewing the rest of the tables for similar issues. Furthermore, we encourage the users to use italics, bold, and bolded italics for greater clarity. They can align all the results to the right, making it evident where differences of magnitude exist.

(7)- Please consider not using brackets to report the standard error but the standard ± notation at all tables reporting the result. Furthermore, ensure all standard errors are reported in the same row as the value they refer to.

(8)- Please make sure acronyms are introduced the first time they are mentioned in the text and used later. E.g., ESG first appears in line 71 but is defined as an acronym at the conclusion (line 396).
